# Confirming the Molecular Basis of the Solvent Extraction of Cadmium(II) Using 2-Pyridyl Oximes through a Synthetic Inorganic Chemistry Approach and a Proposal for More Efficient Extractants

**DOI:** 10.3390/molecules27051619

**Published:** 2022-02-28

**Authors:** Anastasia Routzomani, Zoi G. Lada, Varvara Angelidou, Catherine P. Raptopoulou, Vassilis Psycharis, Konstantis F. Konidaris, Christos T. Chasapis, Spyros P. Perlepes

**Affiliations:** 1Department of Chemistry, University of Patras, 265 04 Patras, Greece; a.routzomani@gmail.com (A.R.); zoilada@iceht.forth.gr (Z.G.L.); varvara_angelidou@hotmail.com (V.A.); 2Institute of Chemical Engineering Sciences (ICE-HT), Foundation for Research and Technology-Hellas (FORTH), Platani, P.O. Box 1414, 265 04 Patras, Greece; 3Institute of Nanoscience and Nanotechnology, NCSR “Demokritos”, Aghia Paraskevi, Attikis, 153 10 Athens, Greece; c.raptopoulou@inn.demokritos.gr; 4Department of Science and High Technology and INSTM, University of Insubria, 22 100 Como, Italy; 5NMR Facility, Instrumental Analysis Laboratory, School of Natural Sciences, University of Patras, 265 04 Patras, Greece

**Keywords:** cadmium(II), coordination chemistry, pyridyl oximes as ligands, single crystal X-ray structures, spectroscopic studies

## Abstract

The present work describes the reactions of CdI_2_ with 2-pyridyl aldoxime (2paoH), 3-pyridyl aldoxime (3paoH), 4-pyridyl aldoxime (4paoH), 2-6-diacetylpyridine dioxime (dapdoH_2_) and 2,6-pyridyl diamidoxime (LH_4_). The primary goal was to contribute to understanding the molecular basis of the very good liquid extraction ability of 2-pyridyl ketoximes with long aliphatic chains towards toxic Cd(II) and the inability of their 4-pyridyl isomers for this extraction. Our systematic investigation provided access to coordination complexes [CdI_2_(2paoH)_2_] (**1**), {[CdI_2_(3paoH)_2_]}_n_ (**2**), {[CdI_2_(4paoH)_2_]}_n_ (**3**) and [CdI_2_(dapdoH_2_**)**] (**4**). The reaction of CdI_2_ and LH_4_ in EtOH resulted in a Cd(II)-involving reaction of the bis(amidoxime) and isolation of [CdI_2_(L’H_2_)] (**5**), where L’H_2_ is the new ligand 2,6-bis(ethoxy)pyridine diimine. A mechanism of this transformation has been proposed. The structures of **1**, **2**, **3**, **4·**2EtOH and **5** were determined by single-crystal X-ray crystallography. The complexes have been characterized by FT-IR and FT-Raman spectra in the solid state and the data are discussed in terms of structural features. The stability of the complexes in DMSO was investigated by ^1^H NMR spectroscopy. Our studies confirm that the excellent extraction ability of 2-pyridyl ketoximes is due to the chelating nature of the extractants leading to thermodynamically stable Cd(II) complexes. The monodentate coordination of 4-pyridyl ketoximes (as confirmed in our model complexes with 4paoH and 3paoH) seems to be responsible for their poor performance as extractants.

## 1. Introduction

Organic matter and heavy toxic metals are the main pollutants of wastewaters, the threat from the latter being more serious [1,2,3,4,5,6]. This is due to the non-biodegradable and non-decomposable nature of the toxic metals, making the development of efficient approaches for their removal and uptake extremely important [7]. Methods in action involve chemical precipitation, microbial treatment, electrodeposition, reverse osmosis, physical/chemical adsorption and solvent extraction [1,8,9,10,11,12]. Solvent extraction is very useful for base metal recovery; the desired metal ions are transferred selectively (after the ore is leached into an aqueous solution) to an H_2_O-immiscible phase with ligands as ion-exchangers, which release an equivalent number of H_3_O^+^ ions back to the aqueous feed solution [13]. For the removal of toxic metal ions from aqueous environments, a special method of solvent extraction, named liquid-liquid extraction, is common [12]. Three types of liquid-liquid extraction are currently used. All three involve the metal-ion association with an organic complexant (extractant) to form a species that is transferred from the aqueous to the organic phase in a two-phase system [12]. In the first type, both the complexant and the metal ion are soluble in the organic phase. In the second type, the complexant and the metal ion source are insoluble in the aqueous phase, the complexation reaction occurs at the interphase region and the metal species is then transferred into the organic phase. In the third type of liquid-liquid extraction, the extractant is soluble in the organic phase (hexane, kerosene, and chloroform for laboratory experiments) and the source of the metal ion is soluble only in the aqueous phase; after the complexation reaction that occurs at the interphase surface, the metal-extractant coordination complex is transferred into the organic phase. The present model study refers to the latter type of liquid-liquid extraction. Most efficient extractants include chelating or even macrocyclic ligands [12]. Today, it is firmly established that an effective extractant should [14]: (a) selectively coordinate to the toxic metal ion having no or very weak affinity for alkali and alkaline metal ions (e.g., Na^+^, Ca^2+^, etc.) which are present at higher concentrations in waste and natural waters; (b) give somewhat thermodynamically stable complex with the toxic metal ion; (c) have fast kinetics with the metal ion to be extracted; (d) resist hydrolysis; and (e) result in a reversible complexation allowing for the easy and complete recovery of the metal ion without destruction or decomposition of the extractant.

The history of cadmium was brilliant in the past [15,16]. Contrary to the past, the present and future of cadmium (both as element and +2 cation) seem dark. Cd is a very toxic metal and is considered as one of the 13 most dangerous metals by the Environmental Protection Agency (EPA) in the U.S. [17,18]. Its ion has generally a short life span and is rendered inert after a period of action. The maximum limit of Cd(II) for humans in H_2_O is 10 mg L^−1^. It is introduced into the environment by metallurgical processes (e.g., Pb-Zn mining) and wastes from electroplating and companies producing pigments, materials for photography and alkaline batteries [1,2,15,19,20,21]. Exposure to Cd(II) causes damages in heart, bones, lungs, and mainly in the kidneys where it is collected affecting their filtering ability [22]. Cadmium(II) compounds are widely classified as carcinogens for humans; most data come from detailed studies which have proven that occupational exposure to Cd(II) sources is closely associated with lung cancer, and possibly to human prostate and renal cancers [23]. Thus, methods for Cd(II) uptake from wastewaters are becoming more and more important.

In the liquid-liquid extraction of toxic Cd(II), several extractants have been used including EDTA derivatives, organophosphorus compounds, dithiocarbamates, crown ethers, calix[4]arene derivatives, 8-quinolols, pyridine carboxamides and pyridyl oximes [12,24]. The stimulus of the present work and a previous publication from our groups [14] was an excellent liquid-liquid extraction study of Cd(II) from aqueous chloride solutions using 1-(2-pyridyl)-tridecane-1-one (2PC12), 1-(2-pyridyl)-pentadecane-1-one (2PC14), 1-(4-pyridyl)-tridecane-1-one (4PC12) and 1-(4-pyridyl)-pentadecane-1-one (4PC14) oximes [25], Figure 1. We are doing a parenthesis here to mention that 2PC12, among other 2-pyridyl ketoximes, has also been successfully used for the liquid-liquid extraction of copper(II) from aqueous solutions containing Cl^−^ ions [26]. Parus, A. et al. examined carefully the influence of extractant, Cd(II) and Cl^−^ concentrations, and the nature of several polar and non-polar organic solvents (diluents) on the extraction efficiency. The study revealed two general experimental facts: (a) Cd(II) is extracted using only 2PC12 and 2PC14. 4PC12 and 4PC14 did not transport complexes with Cd(II) to organic phases. The organic phase employed was CHCl_3_ or hydrocarbons mixed with decan-1-ol. The metal ion was stripped from the loaded organic phase with H_2_O and aqueous NH_3_ solutions; and (b) Based on the effect of the varying concentrations of 2PC12 and 2PC14 on the extraction capability, it was proposed that the chemical species formed and transferred into the organic phase are the neutral complexes [CdCl_2_(2PC12)_2_] and [CdCl_2_(2PC14)_2_]. Using a synthetic coordination chemistry approach [14], we proved that such complexes are capable of existence. The primary goal of the present study was to contribute in understanding the molecular basis of the experimental fact (a) mentioned above. Although the reason of the superior extraction capability of 2PC12, 2PC14 compared to those of 4PC12, 4PC14 might be obvious, i.e., the formation of stable 5-membered chelating rings with the N-donor atoms of the former extractants—which give a thermodynamic stability of the complexes in solution—and the inability of chelating behavior in the case of 4PC12, 4PC14, we were interested in providing synthetic and structural evidence for this working with model complexes. Thus, we used the three isomers of pyridyl aldoximes (2-pyridyl aldoxime, 2paoH; 3-pyridyl aldoxime, 3paoH; 4-pyridyl aldoxime, 4paoH), Figure 1, which gave crystalline complexes. The compounds 2paoH and 4paoH are somewhat satisfactory analogs (albeit not ideal ones) of the extractants 2PC12, 2PC14 and 4PC12, 4PC14, respectively. The main difference is the presence of a long aliphatic (hence hydrophobic) chain in the real extractants instead of the -CHNOH group in the ligands. Cadmium(II) complexes with the ligand 3paoH were also studied. This paper describes results from the synthetic investigation of the general reaction systems CdI_2_/2paoH, CdI_2_/3paoH and CdI_2_/4paoH. For consistency reasons, we used the same inorganic anion of the Cd(II) source; although we originally worked with CdCl_2_·2H_2_O to better simulate the real extraction conditions (e.g., chloride solutions), CdI_2_ gave better crystallinity of the products and their structures were solved through single crystal X-ray crystallography (vide infra).

A secondary goal of this work was to propose types of ligands that might be better extractants than the 2-pyridyl ketoximes 2PC12 and 2PC14. For this reason, we performed reactions of CdI_2_ with 2,6-diacetylpyridine dioxime (dapdoH_2_, Figure 1) and 2,6-diacetylpyridine diamidoxime (LH_4_, Figure 1), to investigate whether the potentially N(pyridyl), {N(oxime)}_2_ tridentate chelating character of these ligands could be realized; in such a case, molecules analogous to dapdoH_2_ and LH_4_ might be better extractants for Cd(II). The ligand LH_4_ possesses two amidoxime moieties which, when introduced in materials (e.g., polymers) give excellent adsorbents for the efficient recovery of Cd(II) from aqueous media [1,21,28].

The spectroscopic studies and the structures of the prepared complexes emphasize the importance of the information emerging from synthetic investigations that otherwise would not have been revealed through exclusively solution studies, while simultaneously shedding light onto the connection of the requisite chemistry with the removal of toxic Cd(II) using pyridyl oxime extractants [19]. The present work can be considered as a continuation of our intense interest in the coordination chemistry of pyridyl aldo/ketoximes and dioximes, and amidoximes/amidodioximes [14,27,29,30,31,32,33,34,35,36,37,38,39,40] and on various aspects of Cd(II) chemistry [14,40,41,42,43,44,45].

## 2. Results and Discussion

### 2.1. Comments on the Syntheses of the Complexes

Since the pH of the aqueous phases during the real extraction experiments was ~4.0 and the extractants were neutral, we avoided adding external bases (e.g., OH^−^, Et_3_N, etc.) which would deprotonate the ligands used. The optimized reaction systems used are conveniently summarized in Figure 2.

The reaction schemes illustrated in Figure 2 deserve synthetic comments. Thus: (i) The use of 1:1 reaction ratio in the case of the CdI_2_/(2,3,4)paoH systems (the complexes were isolated using 1:2 reaction ratios, which coincide with their stoichiometries) does not affect the product identity, leading again to compounds **1**–**3** (microanalytical and IR evidences). (ii) Complexes **2**,**3** can be precipitated from MeCN solutions (IR evidence), but their crystallinity is lower compared to that using EtOH and Me_2_CO as solvents, respectively. (iii) The use of excess dapdoH_2_ in its reaction with CdI_2_ (e.g., a 2:1 dapdoH_2_:CdI_2_ ratio) does not affect the product identity (IR and Raman evidences); the complex was originally isolated using the 1:1 reaction which coincides with its stoichiometry. However, if the reaction solution is concentrated enough (to increase the low yield), the solid product is contaminated with free ligand; and (iv) The study of the 1:1 CdI_2_/LH_4_ reaction system in EtOH provided us with surprising results. The initially precipitated solid (in a yield of ~10%) was analyzed perfectly as [CdI_2_(LH_4_)] (**5a**), as expected. The identity of the product was confirmed by its IR and ^1^H NMR spectra (vide infra). Despite our intense efforts, we could not grow crystals of **5a** for detailed structural characterization. Somewhat to our surprise, an unexpected experimental result was observed during the storage of the filtrate at room temperature; within four days, its color turned slowly pale violet! The pale violet solution was kept in the refrigerator (~5 °C) and X-ray quality, colorless crystals of [CdI_2_(L′H_2_)] (**5**) were grown in a ~40% yield (based on the initially used metal source) within one week. Single-crystal X-ray crystallography revealed that **5** contained the transformed ligand L′H_2_, i.e., the LH_4_ → L′H_2_ transformation had occurred. The free compound L′H_2_ is not known in organic chemistry. We do believe that the observed transformation is CdI_2_-promoted/assisted, since ethanolic solutions of LH_4_ are stable for months. The metal ion-involving reactions of the amidoxime group are well known [46] in the field of the reactivity of coordinated ligands. No analogous reaction has been observed to date in the chemistry of amidoximes. Without any detailed mechanistic studies, we propose the simplified mechanism shown in Figure 3 for the LH_4_ → L′H_2_ transformation. The formation of the L′H_2_ probably involves nucleophilic attack of the solvent to the oxime C atom (Cd(II) coordination to the oxime nitrogen might activate the amidoxime groups towards nucleophilic attack) followed by the reduction in the oxime group by the generated HI, with the simultaneous production of I_2_; the latter justifies the observation of the pale violet color of the reaction solution. In accordance with our proposal, the transformation does not take place in MeCN from which a very small quantity of **5a** is precipitated.

### 2.2. Spectroscopic Characterization in Brief

The complexes were characterized in the solid state by IR and Raman spectroscopies. Representative spectra are shown in Figure 4, Figure 5, Appendix A, Appendix A, Appendix A, Appendix A and Appendix A. The spectra do not exhibit bands that are present in the free ligands 2paoH, 3paoH, 4paoH, dapdoH_2_ and LH_4_, suggesting their purity; if such bands were present, the complexes would have been contaminated with the free ligands used as starting materials. The presence of neutral oxime groups in **1**, **2**, **3** and **4** (well-dried unsolvated sample) is manifested by broad bands (with 2–4 submaxima in the spectra of **1** and **4**) at ~3400 cm^−1^ assigned to *ν*(OH) [38,39]. The bands at 3444 and 3292 cm^−1^ in the IR spectrum of **5** reflect the existence of the imino (=NH) groups in the complex [47]. In **5a**, the bands at 3480 [*ν*_as_(NH_2_)], 3432 [*ν*_s_(NH_2_)] and 3372 [*ν*(OH)] reflect the existence of -NH_2_ and -OH groups supporting the incorporation of coordinated LH_4_ in the complex. The corresponding bands in the free LH_4_ appear at 3484, 3420 and ~3380 cm^−1^. The broadness of the *ν*(OH), *ν*(NH), *ν*_as_(NH_2_) and *ν*_s_(NH_2_) bands, combined with their relatively low wavenumber, are both indicative of hydrogen bonding [38,39]. As expected, the O-H and N-H stretching vibrations are hardly seen in the Raman spectra of the complexes. The peaks at 3076-2922 cm^−1^ are assigned to *ν*(C-H) vibrations [14,48,49]. The in-plane, *δ*(py), and out-of-plane, *γ*(py), deformation vibrations of the 2-pyridyl ring of free paoH (at 627 and 404 cm^−1^, respectively) shift upwards (at 646 and 466 cm^−1^, respectively) in **1** suggesting coordination of the ring-N atom [38]. The *δ*(py) vibration appears as a weak peak in its Raman spectrum [48]. The same trend is observed in the vibrational spectra of the other complexes. For example, the *δ*(py) and *γ*(py) bands are at 658 and 486 cm^−1^, respectively, in the IR spectrum of **3**, while the corresponding vibrations in the spectrum of free 4paoH are located at 640 and 452 cm^−1^, respectively. The *ν*(C=N) vibration of the oxime group(s) in the IR spectra of **1**, **2**, **3**, **4** and **5a** appear as medium to weak bands at 1640, 1624, 1638, 1594 (overlapping with an aromatic stretch) and 1654 cm^−1^, respectively [38]; the corresponding Raman vibrations are assigned [14,48] to the peaks at 1632 (**1**), 1624 (**2**), 1622 (**3**) and 1589 (**4**) cm^−1^. The wavenumbers for **2** and **3** are approximately the same with those of the free ligands 3paoH and 4paoH, respectively. This is strong evidence that the oxime nitrogen does not participate in coordination to Cd(II), a fact that has been confirmed through single-crystal X-ray crystallography (vide infra). The *ν*(C=N) band/peak (IR at 1594 cm^−1^ and Raman at 1638 cm^−1^) for **4** is located at lower wavenumbers compared with those of free dapdoH_2_, suggesting oxime-N coordination. Somewhat to our surprise, the 1626 cm^−1^ band of 2paoH is shifted to a higher wavenumber in the IR spectrum of **1** (1640 cm^−1^) for which the oxime-N coordination has been confirmed (vide infra). This fact, which is not unusual [14], has been interpreted [50] on the basis that some ligands containing a C=N bond (with the carbon atom attached to an aromatic ring) have been shown to undergo a change in the s character of the N lone pair upon coordination; the s character of the N orbital in the C=N bond increases resulting in a greater C=N stretching force constant relative to the free neutral ligands, and this in turn shifts the *ν*(C=N) band in the spectra of the complexes to higher frequencies. An analogous trend is observed in the IR spectrum of **5a** for which coordination of both oxime N atoms is proposed. The medium-intensity IR band at 942 cm^−1^ for free 2paoH is assigned to the *ν*(NO)_oxime_ mode; the corresponding weak Raman peak appears at 950 cm^−1^ [49]. The 942 cm^−1^ band is shifted to a lower wavenumber (932 cm^−1^) in **1**; this shift is due [14,39] to the coordination of the neutral oxime nitrogen. The same trend is observed in the Raman spectrum of the complex where this mode is located at 925 cm^−1^. The assignment of the *ν*(NO)_oxime_ mode for 3paoH, 4paoH, dapdoH_2_ and LH_4_ is not an easy task and any discussion about coordination (or non-coordination) shifts in the spectra of the complexes would be risky.

The complexes were characterized in solution by ^1^H NMR spectroscopy in d_6_-DMSO (Appendix A, Appendix A and Appendix A). The spectrum of free 2paoH displays singlet signals at δ 11.66 and 8.08 ppm assigned to the hydroxyl proton and to the proton attached to the oxime carbon, respectively, and a doublet signal at δ 8.57 ppm assigned to the *α* aromatic proton (i.e., the proton bonded to the carbon next to the pyridyl nitrogen atom). The corresponding signals in the spectrum of **1** appear at δ 10.97, 7.42 and 9.13 ppm. The upfield shift of the oxime carbon-bonded proton and the downfield shift of the *α* proton in the spectrum of the complex both indicate coordination of the two 2paoH N atoms in solution [51], suggesting that the structure of **1** is retained in solution. This is corroborated by the almost zero value of the molar conductivity of the complex in DMSO (Λ_M_ = 4 S cm^2^ mol^−1^ for a 10^−3^ M solution at 25 °C) [52]. Given the stability of **1** in DMSO, we were rather surprised to realize that **4** (which possesses two chelating rings per ligand dapdoH_2_) decomposes in solution! Its ^1^H NMR spectrum in d_6_-DMSO is identical to that of free dapdoH_2_ displaying a simple set of signals at δ 11.51 (-OH, singlet), 7.80 (aromatic protons, multiplet) and 2.25 (-CH_3_, singlet) ppm in the expected 2:3:6 integration ratio. This result, together with the negligible Λ_M_ value in DMSO, indicates a decomposition probably through Equation (1), where x ≥ 4. The Cd^II^-I bond is very stable as suggested by the absence of crystal structures of Cd(II)-DMSO complexes possessing ionic iodides [53]. A rather poor-quality (due to solubility reasons) ^1^H NMR spectrum of **4** in CDCl_3_ is complicated indicating two different solution species, both of which seem to contain coordinated dapdoH_2_. The spectrum of **2** in d_6_-DMSO is identical to that of free 3paoH displaying signals at δ 11.57 (-OH, singlet), 8.21 (-CHNOH) and doublets/multiplets at 8.75, 8.57, 8.00, 7.45 (aromatic protons) ppm in the expected 1:1:4 integration ratio. Similarly, the spectrum of **3** in the same solvent shows signals at δ 11.82 (-OH, singlet), 8.18 (-CHNOH, singlet), and 8.60, 7.55 (aromatic protons) ppm being identical with that of free 4paoH. These data indicate that complexes **2** and **3** decompose in solution releasing 3paoH and 4paoH, respectively. The Λ_M_ values of **2** and **3** in DMSO were very small probably indicating decomposition through Equation (2), x ≥ 4 and a = 2 or 3.
(1)$$[CdI_{2}(dapdoH_{2})]+xDMSO\overset{\mathrm{DMSO}}{\to }[CdI_{2}{\left(DMSO\right)}_{x}]+dapdoH_{2}$$
(2)$${\left\{[CdI_{2}{\left(apaoH\right)}_{2}]\right\}}_{n}+\mathrm{xn}\;\mathrm{DMSO}\overset{\mathrm{DMSO}}{\to }n\;[CdI_{2}{\left(DMSO\right)}_{2}]+2n\;\mathrm{apaoH}\;$$

### 2.3. Description of Structures

The structures of **1**–**3**, **4·**2EtOH and **5** were determined by single-crystal X-ray crystallography. Crystallographic data are gathered in Table 1. Structural plots are shown in Figure 6, Figure 7, Figure 8, Figure 9, Figure 10 and Figure 11 and S9–S12. Selected interatomic distances and angles are listed in Table 2, Table 3, Table 4, Table 5 and Table 6.

Complex [CdI_2_(2paoH)_2_] (1) crystallizes in the monoclinic space group *C*2/*c*. As the complex possesses a 2-fold axis of symmetry passing through Cd1 and bisecting the I1-Cd1-I1**′** angle (Wyckoff position 4e: O, *y*, 1/4), there is 1/2 [CdI_2_(2paoH)_2_] molecule in the asymmetric unit of the cell. The Cd^II^ center forms coordination bonds with two terminal iodo (or iodido) groups (I1, I1 ′), two oxime nitrogen atoms (N1, N1 ′) and two 2-pyridyl nitrogen atoms (N2, N2 ′); the four nitrogen atoms belong to two 1.011 (Harris notation [54]) 2paoH ligands. The coordination polyhedron of the metal ion is a very distorted octahedron, the *cis* and *trans* donor atom-metal ion-donor atom bond angles being in the ranges 66.9(2)–114.8(1) and 145.8(2)–153.6(1) °, respectively. The distorted geometry arises from the small bite angles of the two 5-membered N(oxime)CCN(2-pyridyl)Cd1 rings; these two angles (equal by symmetry) are 66.9(2) °, much smaller than the ideal value of 90 °. The octahedral molecule is the *cis-cis-trans* isomer considering the octahedral positions of the iodo groups, the 2-pyridyl and the oxime nitrogen atoms, respectively. The Cd^II^-N(2-pyridyl) bond length [2.402(5) Å] is smaller than the Cd^II^-N(oxime) one [2.457(6) Å]. There are two intramolecular hydrogen bonds with the uncoordinated oxime oxygen atoms (O1, O1 **′**) as donors and the coordinated iodo groups (I1, I1 **′**) as acceptors, namely O1-H(O1)^…^I1 **′** [and O1 **′**-H(O1 **′**)^…^I1]. Their parameters are: O1^…^I1 **′** = 3.605(5) Å, H(O1)^...^I1 **′** = 2.90 Å and O1-H(O1)^…^I1 **′** = 142.0 °.

Neighboring molecules of 1 interact through π-π stacking interactions involving the 2-pyridyl rings (symmetry operation: −*x* −1/2, −*y* + 1/2, −*z*) forming chains parallel to the [101] crystallographic direction (Appendix A), which are further connected through weak hydrogen bonding interactions creating the 3D architecture of the crystal structure. The distance between the neighboring centrosymmetric 2-pyridyl rings within the chain is 3.83(1) Å.

Compound 2 consists of linear 1D chains. It crystallizes in the monoclinic space group *C*2/*m* and the asymmetric unit contains ¼ of the repeating unit [CdI_2_(3paoH)_2_]. The Cd1 centers sit on Wyckoff positions 2a (0, 0.0, 0) with 2/*m* point group symmetry, with the 2-fold axis being parallel to the *b* axis and the mirror plane cutting it at the *y* = 0.0 point. The atoms of the organic ligand are located on a mirror plane as all occupy the 4i (*x*, 0.0, *z*) Wyckoff positions with point group symmetry *m*. The I1 atoms are also located on 4i-type sites (*x*, 0, *y*) with a point group symmetry *m*, but in this case the mirror plane crosses the *b* axis at the *y* = 0.5 point. The Cd^II^ and I atoms form chains parallel to the *b* axis and the 3paoH ligands are bonded to the metal ions in directions normal to the chain axis. Thus, the Cd^II^ atoms are doubly bridged by two symmetric μ-iodo groups and two monodentate 1.010 3paoH ligands complete a slightly distorted octahedral coordination at each metal. The donor atom of 3paoH is the pyridyl nitrogen. The Cd-I bond lengths in 2 [2.986(1) Å] are larger than in 1 [2.829(1) Å] due to the bridging character of the iodo ligands in the former as compared to their terminal character in the latter. The Cd^II…^Cd^II^ distance is 4.162(1) Å. Neighboring chains in the crystal interact through intermolecular O1-H(O1)^…^I1 hydrogen bonds and form layers parallel to the (110) plane (Appendix A). The metric parameters of the crystallographically unique hydrogen bond O1-H(O1)^…^I1 (*x* + 1/2, *y* + 1/2, *z*) are: O1^…^I1 = 3.528(10) Å, H(O1)^…^I1 = 2.63(3) Å and O-H(O1)^…^I1 = 174(12)°.

The molecular structure and the supramolecular features of 3 are strikingly similar with those of 2. The Cd1 centers sit on the Wyckoff positions 2b (0, 0.5, 0) with 2/*m* point group symmetry; the 2-fold axis is parallel to the *b* axis and the mirror plane cuts it at the *y* = 0.5 point. Again, the atoms of the 1.010 4paoH ligand are located on a mirror plane, as all occupy the 4i (*x*, 0.5, *z*) Wyckoff positions with *m* point group symmetry. The I1 atoms are also located on 4i-type sites (*x*, 0, *y*) with point group symmetry *m*, but in this case the mirror plane crosses the *b* axis at the y= 0.0 point. The Cd^II…^Cd^II^ distances are exactly the same [4.162(1) Å] in 2 and 3. Similarly to the crystal structure of 2, neighboring chains of 3 interact through O(oxime)-H^…^I hydrogen bonds forming layers parallel to the (110) plane (Appendix A). The metric parameters are somewhat different compared to those in 2, due to the different positions of the oxime groups in the pyridyl rings; these are O1-H(O1)^…^I1 (*x* − 1/2, *y* + 1/2, *z*) = 3.501(3) Å, H(O1)**^…^**I1 (*x* − 1/2, *y* + 1/2, *z*) = 2.81(5) Å and O1-H(O1)^…^I1 (*x* − 1/2, *y* + 1/2, *z*) = 150(4) Å.

The crystal structure of 4·2EtOH consists of mononuclear molecules [CdI_2_(dapdoH_2_)] and lattice EtOH molecules in an 1:2 ratio. The asymmetric unit of the cell contains the full complex molecule and the two solvent molecules. Each of the latter interacts (as acceptor) with one “free” (i.e., uncoordinated) oxime oxygen atom (donor) of the dapdoH_2_ ligand (Figure 9, Appendix A). The Cd^II^ atom forms coordination bonds with two terminal iodo ligands (I1, I2), the two oxime nitrogen atoms (N1, N3) and the pyridyl nitrogen atom (N2) of dapdoH_2_. Thus, the organic molecule behaves as a 1.00111 ligand and participates in two 5-membered chelating rings with the metal ion. The Cd^II^-N(pyridyl) bond [2.333(2) Å] is slightly stronger than the Cd^II^-N(oxime) bonds [2.421(2), 2.443(2) Å]. The terminal Cd^II^-I bonds [2.722(1), 2.733(1) Å] are stronger than the corresponding bonds in 1 [2.829(1) Å], and this is due to the lower coordination number of Cd^II^ in 4·2EtOH (five) compared with that in 1 (six). The coordination geometry of Cd^II^ in the complex is extremely distorted, a fact that is primarily attributed to the small bite angles of the two 5-membered N(oxime)CCN(pyridyl)Cd chelating rings; both N(oxime)-Cd-N(pyridyl) coordination angles are 67.5(1)°. The geometry can be either described as a very distorted trigonal bipyramidal one with atoms N1 and N3 defining the axial positions, or as a very distorted square pyramidal one with atoms I1, I2, N1, N3 occupying the basal plane and N2 being at the apical position.

The molecules [CdI_2_(dapdoH_2_)] form pairs through π-π interactions indicated with dashed dark green lines in Figure 10 (the distance between the planes of neighboring, centrosymmetrically-related molecules is 3.81(1) Å, symmetry: −*x*, −*y* + 2, −*z* + 2) and C4-H(C4)-I2 hydrogen bonds. Neighboring pairs interact further through C9-H_C_(C9)^…^O2 hydrogen bonds along the [**1**–**10**] direction and through π-π interactions along the *b* axis, thus forming layers parallel to the (001) plane. The distance between the planes of neighboring, centrosymmetrically-related (−*x*, −*y* + 1, −*z* + 2) molecules along the *b* axis direction is 3.49(1) Å, and this interaction is indicated with dashed light green lines in Figure 10. Metric parameters of the hydrogen bonds are listed in Appendix A.

Compound 5 crystallizes in the orthorhombic space group *Pcnb*. As the complex possesses a 2-fold axis of symmetry passing through Cd1, N2 and C6 atoms (Wyckoff position 4c: 0, ¼, *z*), the asymmetric unit of the cell contains ½ of the [CdI_2_(L′H_2_)] molecule. The 5-coordinate Cd^II^ atom forms coordination bonds with the two terminal iodo groups (I1, I1′), the two imino nitrogen atoms (N1, N1′) and the pyridyl nitrogen atom (N2) of the transformed ligand L’H_2_. Thus, the organic molecule behaves as a 1.00111 ligand participating in two 5-membered chelating rings with the metal center. The terminal Cd^II^-I bonds [2.738(1) Å] are almost identical to those of 4·2EtOH [average 2.727(1) Å], a consequence of the 5-coordination of Cd^II^ in the two complexes. As in 4·2EtOH, the coordination polyhedron of the metal ion in 5 is extremely distorted, a fact primarily arising from the small N1-Cd1-N2 and N1 ′-Cd1-N2 [68.2(1)°] coordination angles of the chelating “parts” of L′H_2_. The polyhedron can be better described as a very distorted trigonal bipyramid with atoms N1 and N1 ′ occupying the axial positions. Neighboring molecules of 5 interact through pairs of C(methyl)-H^…^π interactions forming chains parallel to the *a* axis; neighboring chains interact through C(methyl)-H^…^I hydrogen bonds creating layers parallel to the (001) crystallographic plane (Appendix A, Appendix A). Compound 5 is the first structurally characterized complex of any metal containing the new ligand L′H_2_.

Complexes 2 and 4·2EtOH are the first structurally characterized Cd(II) complexes with 3paoH and dapdoH_2_ ligands, respectively. Compounds 1 and 3 join a small family of structurally characterized Cd(II) complexes of 2paoH and 4paoH [55,56,57,58,59,60], mainly reported by Fonari’s group in a series of excellent crystal engineering studies. The 2paoH complexes reported are [Cd(O_2_CMe)_2_(2paoH)_2_] [50], [Cd_2_(suc)(2paoH)_4_(H_2_O)_2_]](BF_4_)_2_ [56], {[Cd(suc)(2paoH)_2_]}_n_ [56], [Cd(HCO_2_)_2_(2paoH)_2_] [59], {[Cd(1,4-bdc)(2paoH)]·1.5DMF}_n_ [59], {[Cd(SO_4_)(2paoH)(H_2_O)]}_n_ [59], {[Cd(fum)(2paoH)_2_]}_n_ [60], {[Cd(1,3-bdc)(2paoH)]}_n_ [60], where suc^2−^ is the succinate(-2) ligand, 1,4-bdc^2-^ is the 1,4-benzenedicarboxylate(-2) ligand, fum^2−^ is the fumarate(-2) ligand and 1,3-bdc^2-^ is the 1,3-benzedicarboxylate(-2) ligand. In all these complexes, the 2paoH molecule behaves as a N,N′-bidentate chelating (1.011) ligand. The 4paoH Cd(II) complexes reported are {[Cd(mal)(4paoH)(H_2_O)]}_n_ [56], {[Cd(adi)(4paoH)_2_]·DMF}_n_ [56], {[Cd(SO_4_)(4paoH)_2_(H_2_O)_2_]}_n_ [57], [Cd_2_(O_2_CMe)_4_(4paoH)_4_]·4H_2_O [58], [Cd(O_2_CMe)_2_(4paoH)_3_]·3H_2_O [58], {[Cd(1,3-bdc)(4paoH)(H_2_O)_2_]·DMF·H_2_O}_n_ [60], {[Cd(1,4-bdc)(4paoH)_2_(H_2_O)]·DMF}_n_ [60] and {[Cd(1,4-bdc)(4paoH)_2_]·DMF}_n_ [60], where mal^2−^ is the malonate(-2) ligand and adi^2−^ is the adipate(-2) ligand. As in 3, in all of the just mentioned complexes, 4paoH behaves as an N(pyridyl) [1.010] monodentate ligand. It seems that this monodentate coordination mode is the preferable one for Cd(II).

## 3. Experimental Section

### 3.1. Materials and Spectrocopic- Physical Measurements

Experimental manipulations were carried out under aerobic conditions. Deionized water was received from the in-house facility. Solvents and reagents were purchased from Sigma-Aldrich (Tanfrichen, Germany) and Alfa Aesar (Karlsruhe, Germany), and used as received without extra purification. The free ligands 2,6-diacetylpyridine dioxime (dapdoH_2_, Figure 1) and 2,6-pyridyl-diamidoxime (LH_4_, Figure 1) were synthesized by following the procedures published in the literature [61,62]. The products were recrystallized from refluxing EtOH, and their yields were >70%. The purity of the free organic ligands was checked by microanalyses and ^1^H NMR spectroscopy.

Carbon, hydrogen and nitrogen microanalyses were performed by the Instrumental Analysis Center of the University of Patras. FT-IR spectra were recorded using a Perkin-Elmer spectrometer, model 16PC, manufactured by Perkin-Elmer (Waltham, MA, USA); the samples were in the form of KBr pellets prepared under pressure. FT-Raman spectra were obtained in an EQUINOX spectrometer to which a Bruker (D) FRA—106/S component had been attached (Bruker, Karlsruhe, Germany); an R510 diode-pumped Nd:YAG laser at 1064 nm was used for Raman excitation with a maximum laser power of 500 mW on the sample, utilizing an average of 100 scans at 4 cm^−1^ resolution. ^1^H and ^13^C NMR spectra were recorded on a Bruker Avance DPX spectrometer (Bruker AVANCE, Billerica, MA, USA) at resonance frequencies of 400.13 MHz (^1^H) and 100.62 MHz (^13^C); the signals of the solvent (d_6_-DMSO) were used as a reference. Conductivity measurements were performed at room temperature (23–25 °C) in DMSO with a Metrohm-Herisau E-527 bridge (Herisau, Switzerland) and a cell of standard constant; the concentration of the solution was 10^−3^ M.

### 3.2. Preparation of the Complexes

A variety of reaction systems involving various anions of the Cd(II) sources, and different solvent media, reagent ratios, crystallization techniques, reaction times and temperatures were employed before finding the optimized conditions described below.

*[CdI_2_(2paoH)_2_] (1)*: A solution of CdI_2_ (0.055 g, 0.20 mmol) in MeCN (1 mL) was added to a solution of 2paoH (0.049 g, 0.40 mmol) in CH_2_Cl_2_ (3 mL). The resulting colorless solution was stirred for 10 min, filtered and was allowed to slowly evaporate at room temperature. X-ray quality, colorless crystals of the product were obtained within 3 d. The crystals were collected by filtration, washed with cold MeOH (2 × 0.5 mL) and Et_2_O (2 × 2 mL), and dried in vacuo over anhydrous CaCl_2_. Yield: 74%. Anal. Calcd. (%) for C_12_H_12_N_4_CdI_2_O_2_: C, 23.61; H, 1.99; N, 9.18. Found (%): C, 23.24; H, 1.91; N, 9.15. IR (KBr, cm**^−^**^1^): 3324 w, 3314 w, 3304 w, 3228 wb, 3090 w, 3060 w, 1640 m, 1594 s, 1480 s, 1442 s, 1480 s, 1442 s, 1408 s, 1306 m, 1290 s, 1282 sh, 1256 s, 1212 m, 1150 m, 1106 m, 1052 w, 1004 s, 960 m, 932 s, 886 s, 774 s, 742 m, 672 s, 646 m, 600 m, 568 m, 510 s, 466 w. Raman (cm**^−^**^1^): 3054 m, 3003 w, 1562 m, 1642 s, 1632 m, 1611 m, 1570 s, 1550 s, 1498 m, 1468 m, 1427 w, 1406 m, 1304 w, 1243 w, 1222 s, 1212 s, 1171 w, 1099 m, 1048 w, 1017 m, 925 w, 895 w, 772 w, 670 w, 506 w, 403 m, 301 m, 219 m, 137 s, 127 s, 117 s, 97 m, 76 m, 66 w. ^1^H NMR (d_6_-DMSO, δ/ppm): 10.97 (s, 2H), 7.91 (d, 2H), 7.42 (s, 2H), 7.15 (mt, 4H), 6.72 (t, 2H). ^13^C NMR (d_6_-DMSO, δ/ppm): 173.1, 152.3, 149.6, 137.4, 124.6, 120.5. Λ_Μ_ (DMSO, 10^−3^ M, 25 °C) = 4 S cm^2^ mol^−1^.

*{[CdI_2_(3paoH)_2_]}_n_ (2)*: A solution of CdI_2_ (0.110 g, 0.40 mmol) in EtOH (1 mL) was slowly added to a slightly warm (~40 °C) solution of 3paoH (0.098 g, 0.80 mmol) in the same solvent (3 mL). The resulting colorless solution was stirred for 15 min and stored in a closed vial at 5 °C. X-ray quality, colorless crystals of the product were precipitated within 24 h. The crystals were collected by filtration, washed with cold EtOH (0.5 mL) and Et_2_O (2 × 1 mL), and dried in air. The average yield was 65%. Anal. Calcd. (%) for C_12_H_12_N_4_CdI_2_O_2_: C, 23.61; H, 1.99; N, 9.18. Found (%): C, 23.90; H, 1.97; N, 9.35. IR (KBr, cm^−1^): 3488 mb, 1624 w, 1596 w, 1574 w, 1482 m, 1424 m, 1386 w, 1330 w, 1252 s, 1228 sh, 1186 m, 1118 w, 1090 w, 1050 w, 952 s, 936 sh, 882 s, 804 m, 690 s, 642 m, 534 m, 472 mb. Raman (cm^−1^): 3067 m, 1642 s, 1597 m, 1578 m, 1391 w, 1331 w, 1264 w, 1233 w, 1185 w, 1050 w, 1032 m, 884 w, 643 w, 380 w, 292 w, 245 sh. ^1^H NMR (d_6_-DMSO, δ/ppm): 11.57 (sb, 2H), 8.75 (dd, 2H), 8.57 (dd, 2H), 8.21 (s, 2H), 8.00 (mt, 2H), 7.45 (dd, 2H). Λ_Μ_ (DMSO, 10^−3^M, 25 °C) = 6 S cm^2^ mol^−1^.

*{[CdI_2_(4paoH)_2_]}_n_ (3)*: A solution of CdI_2_ (0.055 g, 0.20 mmol) in Me_2_CO (1 mL) was slowly added to a solution of 4paoH (0.049 g, 0.40 mmol) in the same solvent (3 mL). The resulting colorless solution was stirred and stored at −10 °C for 1 month. No solid was noticed and the solution was layered with Et_2_O (2 mL) and allowed to stand undisturbed at room temperature. Slow mixing gave crystals suitable for single-crystal, X-ray crystallography within 12 d. The crystals were collected by filtration, washed with cold EtOH (0.5 mL) and Et_2_O (2 × 1 mL), and dried in air overnight. Yield: 78%. Anal. Calcd. (%) for C_12_H_12_N_4_CdI_2_O_2_: C, 23.61; H, 1.99; N, 9.18. Found (%): C, 23.87; H, 1.90; N, 9.07. IR (KBr, cm^−1^): 3420 sb, 3010 w, 1638 wb, 1610 m, 1505 w, 1498 sh, 1420 m, 1398 m, 1264 s, 1224 sh, 1060 wb, 1010 w, 962 s, 936 m, 888 w, 814 m, 658 m, 564 w, 510 s, 486 m, 404 m. Raman (cm**^−^**^1^): 3071 m, 3059 w, 1622 sh, 1613 s, 1543 w, 1399 w, 1327 w, 1269 w, 1240 w, 1225 w, 1206 m, 1013 m, 891 w, 668 w, 291 w, 239 w. ^1^H NMR (d_6_-DMSO, δ/ppm): 11.82 (s, 2H), 8.60 (dd, 4H), 8.18 (s, 2H), 7.55 (dd, 4H). Λ_M_ (DMSO, 10^−3^ M, 25 °C) = 5 S cm^2^ mol^−1^.

*[CdI_2_(dapdoH_2_)]·2EtOH (4·2EtOH)*: A solution of CdI_2_ (0.055 g, 0.20 mmol) in EtOH (1 mL) was added to a slurry of dapdoH_2_ (0.039 g, 0.20 mmol) in the same solvent (6 mL). The resulting pale yellow suspension was filtered to remove a very small quantity of the ligand. The filtrate was stirred for 2–3 min and stored in a closed vial at room temperature. X-ray quality, colorless crystals of the product were formed within 24 h which were collected by filtration, washed with cold EtOH (0.5 mL) and Et_2_O (2 × 1 mL), and dried in vacuo over anhydrous CaCl_2_. Yield: 28%. The sample was analyzed without lattice EtOH molecules. Anal. Calcd. (%) for C_9_H_11_N_3_CdI_2_O_2_: C, 19.32; H, 1.99; N, 7.51. Found (%): C, 19.35; H, 2.08; N, 8.11 IR (KBr, cm^−1^): 3422 s, 3178 mb, 3084 w, 3036 w, 2922 w, 1594 m, 1472 m, 1364 m, 1316 w, 1292 m, 1264 m, 1194 w, 1156 w, 1130 w, 1038 s, 950 m, 830 w, 808 m, 732w, 690 m, 654 w, 554 m, 508 w, 428 w. Raman (cm^−1^): 3076 m, 3065 sh, 2932 m, 1638 m, 1589sh, 1564 s, 1476 sh, 1464 w, 1429 w, 1358 m, 1314 w, 1262 w, 1129 w, 1015 s, 832 w, 760 w, 656 w, 448 w. ^1^H NMR (d_6_- DMSO, δ/ppm): 11.51 (s, 2H), 7.80 (mt, 3H), 2.25 (s, 6H). Λ_Μ_ (DMSO, 10^−3^ M, 25 °C) = 8 S cm^2^ mol^−1^.

*[CdI_2_(L′H_2_)] (5)*: A solution of CdI_2_ (0.055 g, 0.20 mmol) in EtOH (1 mL) was added to a solution of LH_4_ (0.039 g, 0.20 mmol) in the same solvent (7 mL). The resulting suspension was filtered to remove a small quantity of a precipitated material (5a). The filtrate was stirred for 2–3 min and stored in a closed vial at room temperature for 4d. The color of the reaction solution (filtrate) turned gradually to pink-pale violet during this time. This pink solution was placed at 5 °C, and X-ray quality, colorless crystals (from a pink solution!) were grown within a period of 7–8 d. The crystals were collected by filtration, washed with cold EtOH (0.5 mL) and Et_2_O (2 × 1 mL), and dried in air overnight. Yield: 41%. The sample 5a was analyzed as [CdI_2_(LH_4_)]. Anal. Calcd. (%) for C_7_H_9_N_5_CdI_2_O_2_: C, 14.97; H, 1.62; N, 12.48. Found (%): C, 14.60; H, 1.67; N, 12.63. IR spectrum (KBr, cm^−1^) for 5a: 3480 s, 3432 sb, 3372 sb, 3070 wb, 1654 s, 1604 sh, 1582 s, 1484 m, 1402 m, 1340 s, 1214 w, 1155 w, 1105 w, 1076 s, 1018 m, 986 s, 902 m, 810 s, 732 m, 696 s, 656 m, 522 mb, 460 w, 446 w. ^1^H NMR spectrum (d_6_-DMSO, δ/ppm) for 5a: 9.85 (s, 2H), 7.81 (mt, 3H), 6.32 (sb, 4H). Analytical data for the product 5: Anal. Calcd. (%) for C_11_H_15_N_3_CdI_2_O_2_: C, 22.49; H, 2.58; N, 7.15. Found (%): C, 28.93; H, 2.71; N, 7.07. IR spectrum (KBr, cm^−1^) for 5: 3444 mb, 3292 mb, 2984 m, 1648 m, 1578 s, 1458 m, 1404 m, 1378 m, 1344 s, 1290 m, 1198 m, 1146 s, 1106 m, 1018 m, 930 w, 838 m, 796 m, 748 m, 696 w, 660 m, 434 w. The Λ_Μ_ values of both 5 and 5a in DMSO are ~5 S cm^2^ mol^−1^.

### 3.3. Single-Crystal X-ray Crystallography

Colorless crystals of 1, 2, 3, 4·2EtOH and 5 were taken from the mother liquor and immediately cooled to 160 (1, 3, 4·2EtOH, 5) or 170 (2) K. Diffraction data were collected on a Rigaku R-AXIS Image Plate diffractometer (Rigaku Americas Corporation, The Woodlands, TX, USA; European Department at Karlsruhe, Germany) using graphite-monochromated Cu Ka (1, 2) or Mo Ka (3, 4·2EtOH, 5) radiations. Data collection (ω-scans) and processing (cell refinement, data reduction, and empirical/numerical absorption correction) were performed using the CrystalClear program package [63]. The structures were solved by direct methods using SHELΧS, ver. 2013/1 [64] and refined by full-matrix least-squares techniques on *F*^2^ with SHELXL, ver. 2014/6 [65]. Most H atoms were introduced at calculated positions as riding on their corresponding bonded atoms. All non-H atoms were refined anisotropically. Plots of the structures were drawn using the Diamond 3 program package [66].

Crystallographic data have been deposited with the Cambridge Crystallographic Data Center, Nos 2142693 (1), 2142694 (2), 2142695 (3), 2142696 (4·2EtOH) and 2142697 (5). Copies of the data can be obtained free of charge upon application to CCDC, 12 Union Road, Cambridge, CB2 1EZ, UK: Tel.: +(44)-1223-762910; Fax: +(44)-1223-336033; E-mail: deposit@ccdc.cam.ac.uk.

## 4. Conclusions in Brief

According to our opinion, the important chemical messages of this work are: (a) The reported complexes enrich the coordination chemistry of 2-pyridyl oximes, especially that with Cd(II). (b) The molecular structures and supramolecular features of the complexes are interesting; and (c) The interesting Cd(II)-assisted/promoted LH_4_ → L’H_2_ transformation has been observed for the first time and this is a new, welcome example in the area of the reactivity of coordinated amidoximes.

From the viewpoint of the solvent extraction of toxic Cd(II) using 2-pyridyl oximes as extractants (which stimulated the present efforts) our inorganic chemistry approach has firmly confirmed the molecular basis of the excellent extraction ability of 2PC12 and 2PC14, and the poor one for 4PC12, 4PC14. With the drawbacks mentioned in Introduction, the very good extraction ability of the 2-pyridyl ketoximes can be attributed to the chelating nature of the extractants as structurally proven by the 2paoH-containing complex 1; this chelating behavior results in thermodynamically stable complexes of Cd(II) with the extractant, favoring this process. The monodentate coordination of 4-pyridyl ketoximes (as structurally proven in the 4paoH-containg compound 3) seems to be responsible for the poor performance of 4PC12 and 4PC14. In an analogous manner (as proven in the 3paoH-conataining compound 2), extractants such as 3PC12 and 3PC14 (i.e., those containing the oxime group at position 3 of the pyridyl ring; Figure 1) are predicted to disfavor the extraction process; such “real” experiments are not available.

We tentatively propose that the structurally established tridentate chelating character of dapdoH_2_ towards Cd(II) in complex 4·2EtOH and the presumable such behavior of LH_4_ in complex 5a could be exploited to develop extractants consisting of one pyridyl ring and two oxime groups that contain long alkyl components at the 2- and 6-positions, or even to carry out experiments with the bis(amidoxime) compound LH_4_ (or better with derivatives containing aliphatic substituents on the pyridyl ring to ensure good solubility in non-polar organic solvents). The decomposition of 4 in DMSO (as evidenced by its ^1^H NMR spectrum in d_6_-DMSO, see Section 2.2) should not be disappointing, since this solvent is an efficient donor forming complexes with Cd(II) and favoring decomposition; the presence of coordinated dapdoH_2_-containing species in CDCl_3_ (albeit evidenced by poor quality ^1^H NMR spectra) is a good sign towards the use of 2,6-pyridyl dioximes as extractants for toxic Cd(II). The scientific community dealing with solvent extraction experiments might obtain good results working with tridentate dioximes.

## Figures and Tables

**Figure 1 molecules-27-01619-f001:**
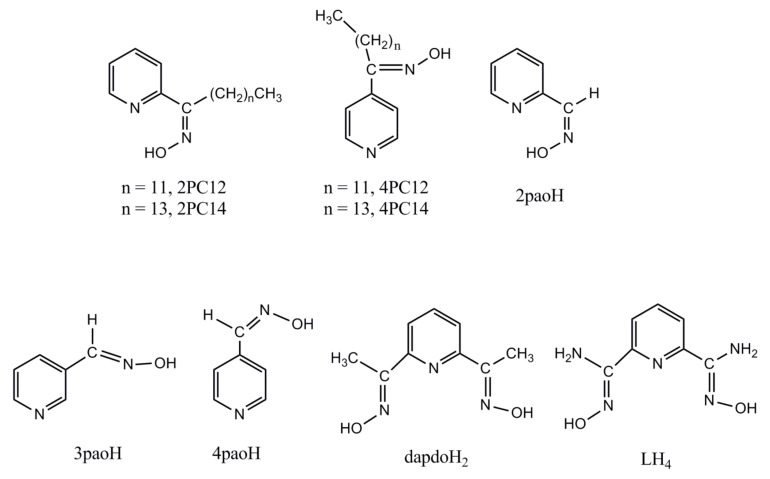
Structural formulae of the extractants and the ligands discussed in the text, and their abbreviations; the presence of four hydrogens in the abbreviation of 2,6-pyridyl diamidoxime (LH_4_) implies the known in the literature possibility of the single deprotonation of the -NH_2_ group [27].

**Figure 2 molecules-27-01619-f002:**
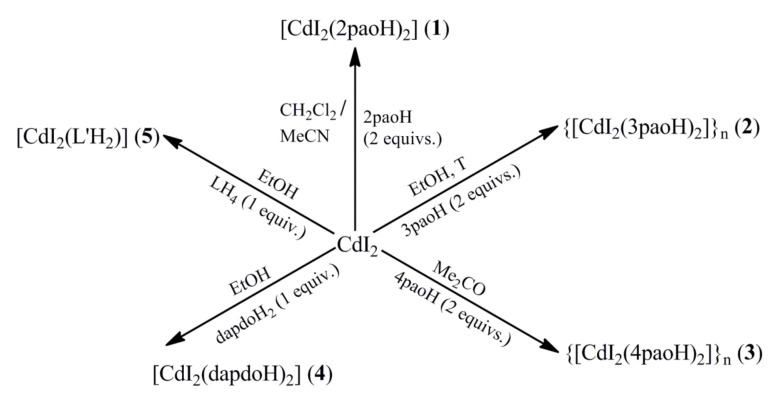
Optimized experimental reaction conditions that lead to Cd(II) complexes (**1**–**5**); the lattice EtOH molecules in the dapdoH_2_ complex are not shown. The structural formula of the ligand L′H_2_ and a possible mechanism of the LH_4_ → L′H_2_ transformation are presented in Figure 3 (vide infra).

**Figure 3 molecules-27-01619-f003:**
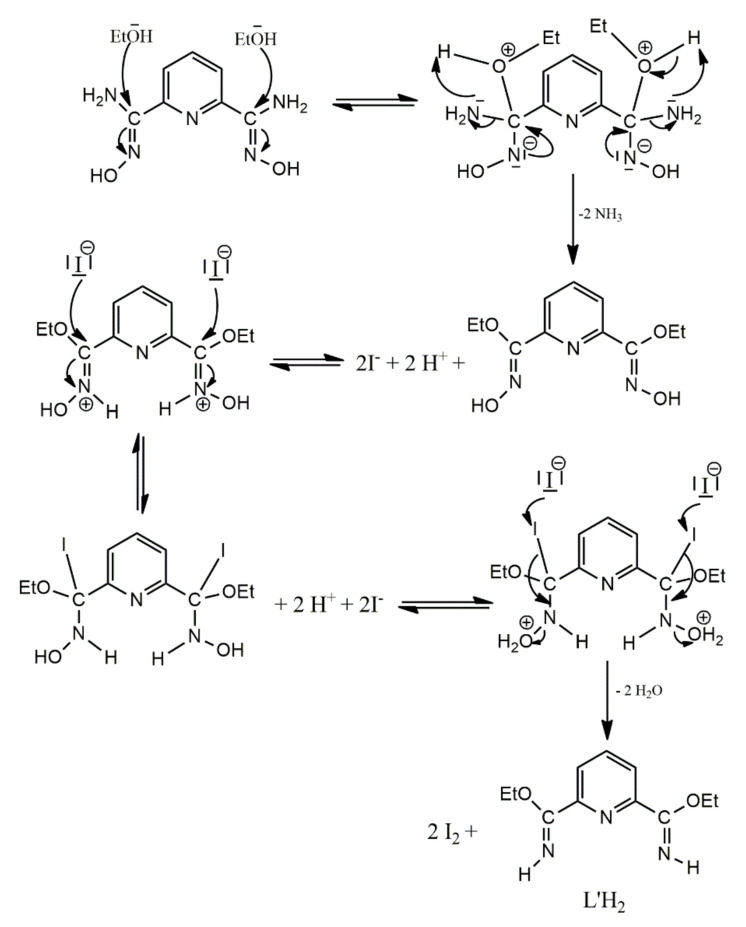
A simplified mechanistic proposal for the CdI_2_-promoted/assisted LH_4_ → L′H_2_ transformation. The I^−^ ions come from the metal source, while the H^+^ ions (as a matter of fact H_3_O^+^) from the first step of hydrolysis/solvolysis of [Cd(H_2_O)_6_]^2+^/[Cd(EtOH)_6_]^2+^; the water is contained in the organic solvent which is not anhydrous.

**Figure 4 molecules-27-01619-f004:**
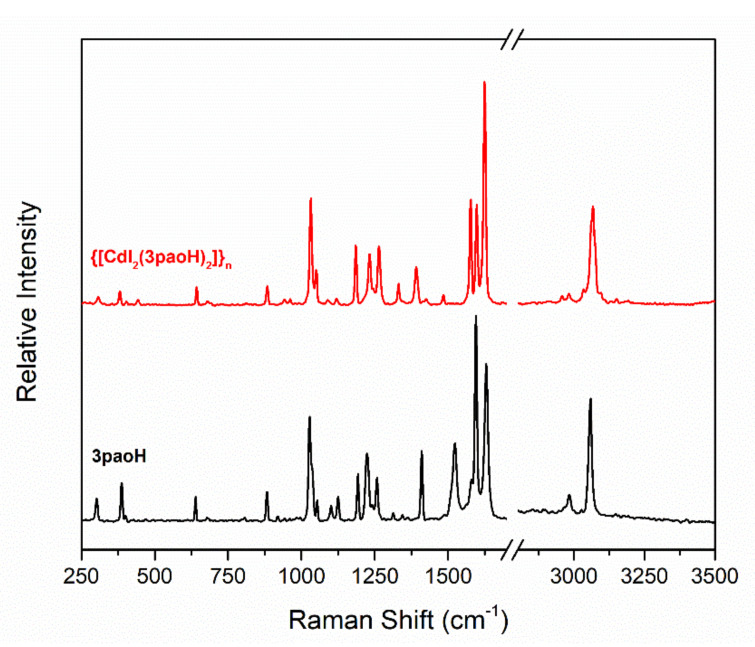
The FT-Raman spectrum of the ligand 3paoH (black line) and the {[CdI_2_(3paoH)_2_]}_n_ (**2**) complex (red line).

**Figure 5 molecules-27-01619-f005:**
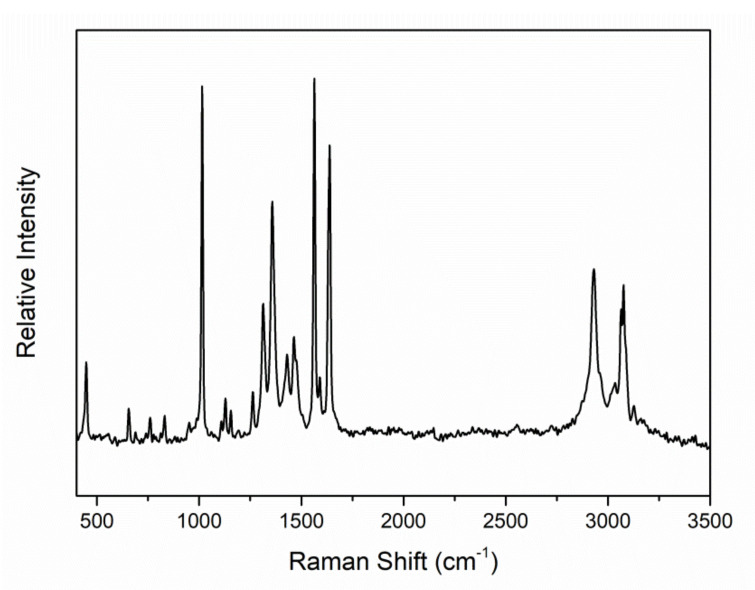
The FT-Raman spectrum of a well-dried (i.e., without lattice EtOH) sample of [CdI_2_(dapdoH_2_)] (**4**).

**Figure 6 molecules-27-01619-f006:**
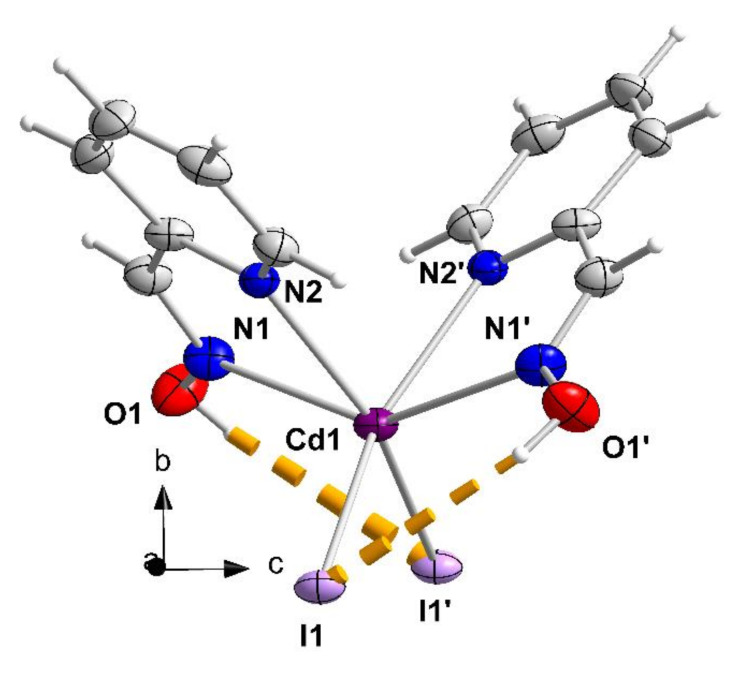
The structure of the [CdI_2_(2paoH)_2_] molecule that is present in **1**. The thermal ellipsoids are presented at the 50% probability level. Only the metal, the donor atoms and the oxime oxygen atoms are numbered. The dashed lines indicate intramolecular hydrogen bonds. Symmetry operation: (′) = *x*, *y*, −*z* + 1/2.

**Figure 7 molecules-27-01619-f007:**
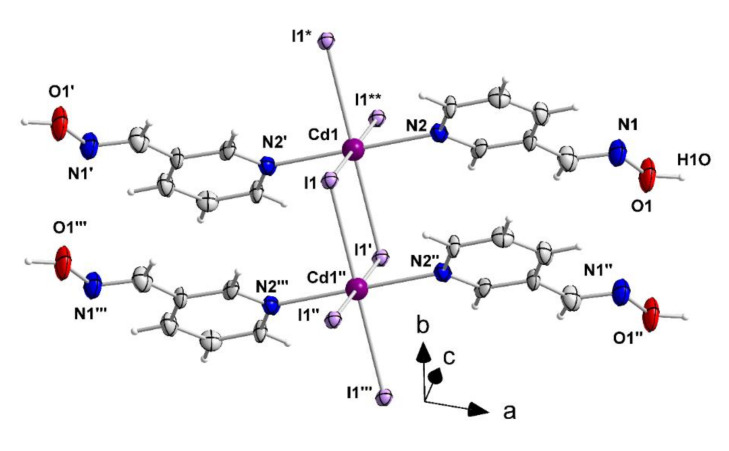
A small portion of the 1D chain that is present in the crystal structure of {[CdI_2_(3paoH)_2_]}_n_ (**2**). The thermal ellipsoids are presented at the 50% probability level. Only the Cd^II^ atoms, the donor atoms and the N, O atoms of the oxime groups are numbered. Symmetry operations: (′)= −*x*, *y*, −*z*; (′′) = *x*, *y* − 1, *z*; (′′′) = −*x*, *y* − 1, −*z*; (*) = *x*, *y* + 1, *z*; (**) = −*x*, *y* + 1, −*z*.

**Figure 8 molecules-27-01619-f008:**
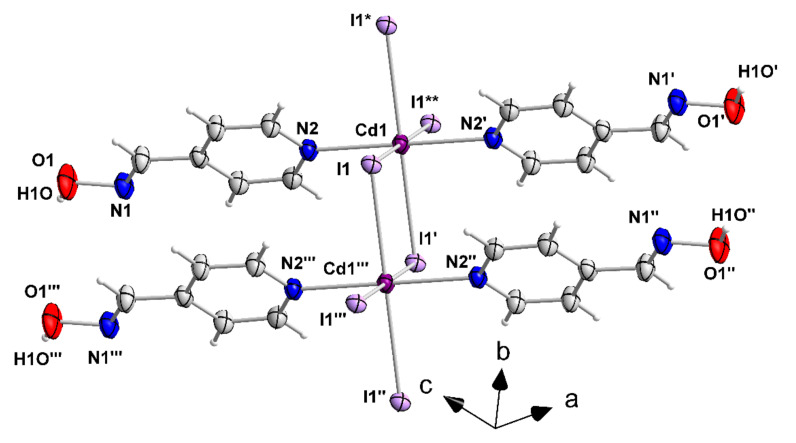
A small portion of the 1D chain that is present in the crystal structure of {[CdI_2_(4paoH)_2_]}_n_ (**3**). The thermal ellipsoids are presented at the 50% probability level. Only the Cd^II^ atoms, the donor atoms and the N, O atoms of the oxime groups are numbered. Symmetry operations: (′) = −*x*, *y*, −*z*; (′′) = −*x*, *y* − 1, −*z*; (′′′) = *x*, *y* − 1, *z*; (*) = *x*, *y* + 1, *z*; (**) = −*x*, *y* + 1, −*z*.

**Figure 9 molecules-27-01619-f009:**
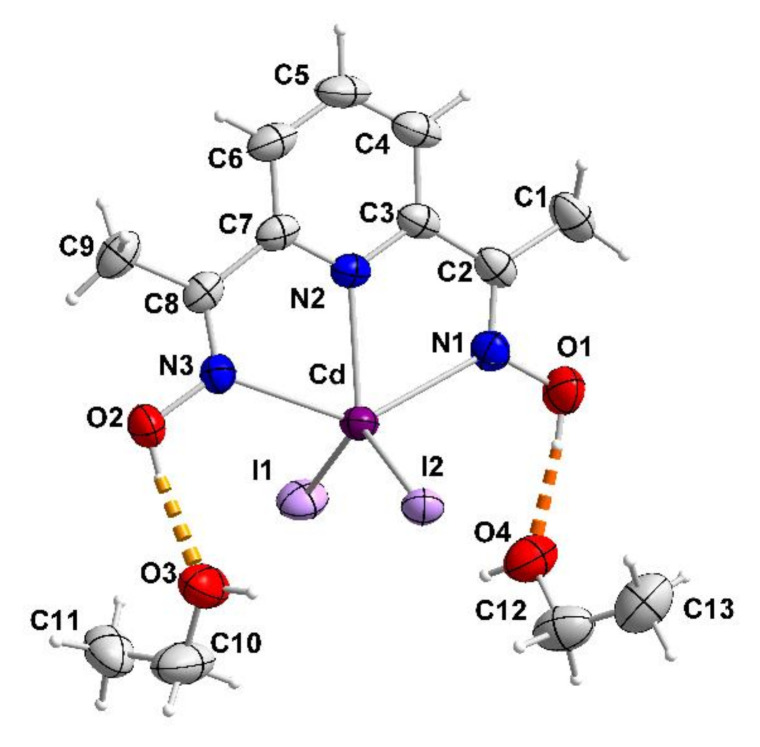
The molecules [CdI_2_(dapdoH_2_)] and EtOH that are present in the crystal structure of 4·2EtOH. The thermal ellipsoids are presented at the 50% probability level. The thick dashed orange lines indicate hydrogen bonds between the oxime groups and the lattice EtOH molecules.

**Figure 10 molecules-27-01619-f010:**
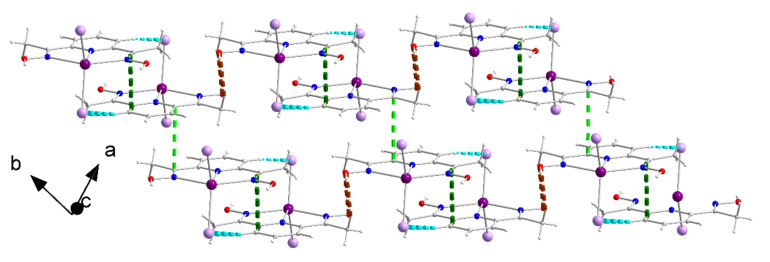
Layers of molecules [CdI_2_(dapdoH_2_)] parallel to the (001) plane in the crystal structure of 4·2EtOH. The thick dashed dark and light green lines represent π-π interactions. The thick dashed cyan and brown lines represent the C4-H(C4)-I2 and C9-H_C_(C9)^…^O2 hydrogen bonds, respectively. The hydrogen bonds involving the lattice EtOH molecules have not been drawn for clarity reasons; for more details see the above text and Appendix A.

**Figure 11 molecules-27-01619-f011:**
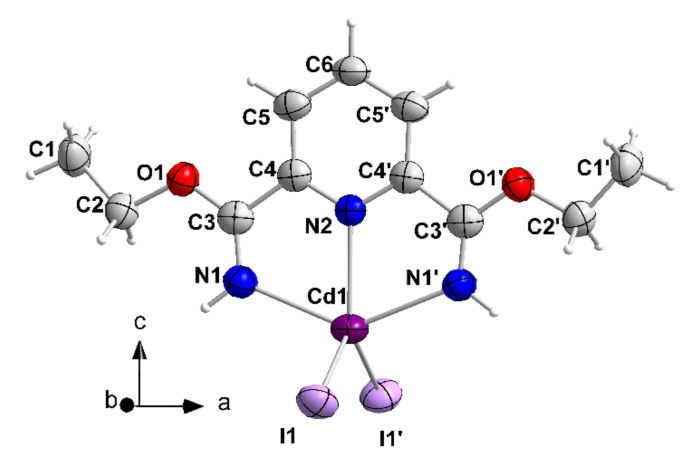
The structure of [CdI_2_(L′H_2_)] molecule that is present in 5. The thermal ellipsoids are presented at the 50% probability level. Symmetry operation: (′) = −*x*, −*y* + 3/2, *z*.

**Table 1 molecules-27-01619-t001:** Crystallographic data and refinement parameters for the structures of **1**, **2**, **3**, **4·**2EtOH and **5**.

Parameter	[CdI_2_(2paoH)_2_] (1)	{[CdI_2_(3paoH)_2_]}_n_ (2)	{[CdI_2_(4paoH)_2_]}_n_ (3)	[CdI_2_(dapdoH_2_)]·2EtOH (4·2EtOH)	[CdI_2_(L′H_2_)] (5)
Empirical formula	C_12_H_12_CdI_2_N_4_O_2_	C_12_H_12_CdI_2_N_4_O_2_	C_12_H_12_CdI_2_N_4_O_2_	C_13_H_23_CdI_2_N_3_O_4_	C_11_H_15_CdI_2_N_3_O_2_
Formula weight	610.46	610.46	610.46	651.54	587.46
Crystal system	monoclinic	Monoclinic	monoclinic	triclinic	orthorhombic
Space group	*C*2/*c*	*C*2/*m*	*C*2/*m*	*P* $\bar{1}$	*Pcnb*
Color	colorless	Colorless	colorless	colorless	colorless
Crystal size, mm	0.41 × 0.20 × 0.17	0.21 × 0.21 × 0.06	0.23 × 0.13 × 0.05	0.35 × 0.18 × 0.13	0.35 × 0.19 × 0.08
*a*, Å	7.9245(2)	25.0595(11)	24.8525(7)	8.0032(19)	6.4986(3)
*b*, Å	13.7155(2)	4.1616(2)	4.1618(10)	9.824(2)	15.0095(6)
*c*, Å	15.6259(2)	7.9660(4)	8.1913(3)	14.215(3)	17.7360(7)
*α,* °	90.0	90.0	90.0	81.439(6)	90.0
*β*, °	101.35(1)	98.413(1)	98.223(1)	78.146(5)	90.0
*γ*, °	90.0	90.0	90.0	74.771(6)	90.0
Volume, Å^3^	1665.13(8)	821.82(7)	838.52(4)	1050.0(4)	1729.98(13)
Z	4	2	2	2	4
Temperature, K	160	170	160	160	160
Radiation, Å/2*θ*_max_	Cu Kα (1.54178)/130.0	Cu Kα (1.54178)/129.8	Mo Kα (0.71073)/54.0	Mo Kα (0.71073)/54.0	Mo Kα (0.71073)/54.0
Calculated density, g·cm^−3^	2.435	2.467	2.418	2.061	2.256
Absorption coefficient, mm^−1^	39.71	40.23	4.99	4.00	4.83
Number of measured, independent, and observed [*I* > 2*σ*(*I*)] reflections	10,738, 1385, 1333	5658, 718, 661	10,187, 1042, 988	32,407, 4564, 4189	35,910, 1875, 1753
Number of parameters	98	65	65	216	89
Final *R* indices [*I* > 2*σ*(*Ι*)] ^a^	*R*_1_ = 0.0384, *w_R_*_2_ = 0.0902	*R*_1_ = 0.0499, *w_R_*_2_ = 0.1079	*R*_1_ = 0.0166, *w_R_*_2_ = 0.0398	*R*_1_ = 0.0224, *w_R_*_2_ = 0.0570	*R*_1_ = 0.0337, *w_R_*_2_ = 0.0707
Goodness-of-fit on *F*^2^	1.06	1.10	1.05	1.02	1.11
Largest differences peak and hole (e Å^−3^)	1.28/−1.66	1.22/−1.20	0.66/−0.52	0.53/−0.74	0.50/−0.84

^a ^*R*_1_ = Σ(|*F*_o_| − |*F*_c_|)/Σ(|*F*_o_|), *wR*_2_ = {Σ[*w*(*F*_o_^2^ − *F*_c_^2^)^2^]/Σ[*w*(*F*_o_^2^)^2^]}^1/2^, *w* = 1⁄[*σ*^2^(*F*_o_^2^) + (α*P*)^2^ + *b**P*], where *P =* [max (*F*_o_^2^,0) + 2*F*_c_^2^]/3. (α = 0.0029 and *b* = 7.5036 for **1**; α = 0.0519 and *b* = 2.3524 for **2**; α = 0.0232 and *b* = 0.8669 for **3**; α = 0.0281 and *b* = 1.0643 for **4·**2EtOH; α = 0.0151, *b* = 7.8149 for **5**).

**Table 2 molecules-27-01619-t002:** Selected bond lengths (Å) and angles (°) for the complex [CdI_2_(2paoH)_2_] (**1**) ^a^.

Bond Lengths (Å)		Bond Angles (°)	
Cd1-I1	2.829(1)	I1-Cd1-I1 ′	103.4(1)
Cd1-N1	2.457(6)	I1-Cd1-N1	114.8(1)
Cd1-N2	2.402(5)	I1-Cd1-N1 ′	86.8(1)
C1-N1	1.253(8)	I1-Cd1-N2	89.8(1)
N1-O1	1.398(7)	I1-Cd1-N2 ′	153.6(1)
		N1-Cd1-N1 ′	145.8(2)
		N1-Cd1-N2	66.9(2)
		N1-Cd1-N2 ′	88.2(2)
		N2-Cd1-N2 ′	87.8(2)

^a^ Symmetry code: (′) = −*x*, *y*, −*z* + 1/2. Atoms C1 and C1′, not labelled in Figure 6, are the oxime carbon atoms of coordinated 2paoH.

**Table 3 molecules-27-01619-t003:** Selected interatomic distances (Å) and angles (°) for the polymeric compound {[CdI_2_(3paoH)_2_]}_n_ (**2**) ^a^.

Interatomic Distances (Å)		Interatomic Angles (°)	
Cd1-N2/N2 ′	2.353(9)	N2-Cd1-N2 ′ = I1-Cd1-I1 ′′ = I1 ′-Cd1-I1 *	180.0(1)
Cd1-I1/I1 ′′	2.986(1)	I1-Cd1-I1 ′ = I1 *-Cd1-I1 **	91.6(1)
Cd1^…^Cd1 ′′	4.162(1)	I1-Cd1-I1 * = I1 ′-Cd1-I1 **	88.4(2)
N1-O1 = N1 ′-O1 ′ = N1 ′′-O1 ′′ = N1 ′′′-O1 ′′′	1.423(14)	I1 ′-Cd1-N2 = I1-Cd1-N2 ′ = I1 *-Cd1-N2 ′ = I1 **-Cd1-N2	90.9(2)
		I1-Cd1-N2 = I1 ′-Cd1-N2 ′ = I1 *-Cd1-N2 = I1 ′′-Cd1-N2 ′	89.1(2)
		Cd1-I1-Cd1 ′′	88.4(2)

^a^ Symmetry codes: (**′**) = −*x*, *y*, −*z*; (**′′**) = *x*, *y* − 1, *z*; (**′′′**) = −*x*, *y* − 1, −*z*; (*) = *x*, *y* + 1, *z*; (**) = −*x*, *y* + 1, −*z*.

**Table 4 molecules-27-01619-t004:** Selected interatomic distances (Å) and angles (°) for the polymeric compound {[CdI_2_(4paoH)_2_]}_n_ (**3**) ^a^.

Interatomic Distances (Å)		Interatomic Angles (°)	
Cd1-N2/N2 ′	2.353(2)	N2-Cd1-N2 ′ = I1-Cd1-I1 ′′ = I1 ′-Cd1-I1 *	180.0(1)
Cd1-I1/I1 ′/I1 */I1 **	2.991(1)	I1-Cd1-I1 ′ = I1 *-Cd1-I1 ′′	91.8(1)
Cd1^…^Cd1 ′′′	4.162(1)	I1-Cd1-I1 * = I1 ′-Cd1-I1 ′′	88.2(1)
N1-O1=N1 ′-O1 ′	1.403(4)	I1-Cd1-N2 ′ = I1 ′-Cd1-N2 = I1 *-Cd1-N2 ′ = I1 **-Cd1-N2	90.7(1)
		I1-Cd1-N2 = I1 ′-Cd1-N2 ′ = I1 *-Cd1-N2 = I1 **-Cd1-N2 ′	89.3(1)
		Cd1-I1-Cd1 ′′′	88.2(1)

^a^ Symmetry codes: (**′**) = −*x*, *y*, −*z*; (**′′**) = −*x*, *y* − 1, −*z*; (**′′′**) = *x*, *y* − 1, *z*; (*) = *x*, *y* + 1, *z*; (**) = −*x*, *y* + 1, −*z*.

**Table 5 molecules-27-01619-t005:** Selected bond lengths (Å) and angles (°) for complex [CdI_2_(dapdoH_2_)]·2EtOH (**4·**2EtOH).

Bond Lengths (Å)		Bond Angles (°)	
Cd-I1	2.722(1)	I1-Cd-I2	126.7(1)
Cd-I2	2.733(1)	I1-Cd-N1	100.2(1)
Cd-N1	2.421(2)	I1-Cd-N2	117.8(1)
Cd-N2	2.333(2)	I1-Cd-N3	98.8(1)
Cd-N3	2.443(2)	I2-Cd-N1	101.0(1)
C2-N1	1.279(4)	I2-Cd-N2	115.4(1)
N1-O1	1.380 (3)	I2-Cd-N3	99.6(1)
C8-N3	1.275(3)	N1-Cd-N2	67.5(1)
N3-O2	1.384(3)	N1-Cd-N3	134.9(1)
		N2-Cd-N3	67.5(1)

**Table 6 molecules-27-01619-t006:** Selected bond lengths (Å) and angles (°) for complex [CdI_2_(L′H_2_)] (**5**) ^a^.

Bond Lengths (Å)		Bond Angles (°)	
Cd1-I1	2.738(1)	I1-Cd1-I1 ′	119.4(1)
Cd1-N1	2.423(4)	I1-Cd1-N1	100.7(1)
Cd1-N2	2.344(5)	I1-Cd1-N2	120.3(1)
C3-N1	1.268(6)	I1-Cd1-N1 ′	100.9(1)
C3-O1	1.335(6)	N1-Cd1-N2	68.2(1)
O1-C2	1.451(6)	N1-Cd1-N1 ′	136.4(1)
C1-C2	1.489(7)		

^a^ Symmetry code: (′) = −*x*, −*y* + 3/2, *z*.

## Data Availability

Not applicable.

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
