# Peer review of "Confirming the Molecular Basis of the Solvent Extraction of Cadmium(II) Using 2-Pyridyl Oximes through a Synthetic Inorganic Chemistry Approach and a Proposal for More Efficient Extractants"

_molecules, 2022, doi:10.3390/molecules27051619_

Round 1

Reviewer 1 Report

The topic of the manuscript is very interesting and concerning the recent developments in the inorganic chemistry field. This manuscript presents a thorough investigation on the synthetic and structural evidences of Cd(II) complexes with five different ligands (2,3,4-paoH, dapdoH2, and LH4). The authors synthesized these complexes and characterized them with FTIR, NMR, and XRD. These compounds are in fact very interesting.  I am in agreement with the authors however, I have a few comments and suggestions that will hopefully improve the manuscript if considered by the authors.

The authors wrote that they checked the purity of compounds with 1H NMR however, some of their complexes 2, 3, and 4 were decomposed in the D6-DMSO. It is then unclear how they check the purity of complexes, 2, 3, and 4? I suggest authors to provide values of %purity of their compounds in the experimental section. Moreover, in lines 260-261, they wrote …’The spectra do not exhibit bands that are present in the free ligands 2paoH, 3paoH, and 4-paoH, dapdoH2, and LH4, suggesting their purity’…. Authors need to clarify their how FTIR spectra confirms purity?

It is hard to follow what authors wants to convey in lines 213 - 214, ‘The use of 1:1 reaction ratio in the case of the CdI2/(2,3,4-paoH) systems does not affect the product identity’. Can they explain what is the product identity? In the same section, in lines 219 - 220, they wrote that the ratio of 2:1 dapdoH2:CdI2) was used while Figure 2 displayed 1:1 ratio between dapdoH2:CdI2 for formation of complex 4. I suggest authors to clarify this and write equivalent of CdI2 used in the Figure 2, perhaps below CdI2.

In lines, 292 - 294, it is unclear which band are they talking. I suggest authors to write band value in cm-1 and explain how much in numbers cm-1 it is decreased.

Figure 3 has very poor resolution. I suggest authors to increase the resolution and the font size in Figure 3.

In Figure 4, I suggest authors to add FT-Raman spectra of free ligand 3paoH so it is easier for readers to see how FT-Raman spectra changes from non-complexed to the complexed structure.

In the XRD structural characterization section, as the structure of compounds 2 and 3 are very similar so I suggest authors to compress their discussion and explain them together.

In line 188 of experimental section, I suggest authors to give average yield of complex 2 instead of range (60 – 70%) which is currently given.

Moreover, conclusion section need to be revised thoroughly. It should be described briefly and scientifically.

The paper is very tedious to read, sometimes non-scientific language is used at several places, and extensive editing of English language and style is recommended. At some places conclusions are merely based on assumptions and not supported by the results.

Few examples are identified here but Language needs to extensively improved,

 line 133, ‘The polish group’. I suggest authors to replace it with the name of authors.

Line 211, …’Figure 2 characteristic deserve synthetic comments’….

Line 230, A peculiar event took place

Line 232, pink-pale violet! I suggest authors to describe this color in a better way so it is easier for readers to follow.

Lines 311-313, 345-347 needs to written more precisely so it is easier to follow.

Line 145 in the experimental section, ‘synthetic manipulations’….

Line 264, conclusion section, welcome case in the area….

Author Response

Reviewer # 1

The topic of the manuscript is very interesting and concerning the recent developments in the inorganic chemistry field. This manuscript presents a thorough investigation on the synthetic and structural evidences of Cd(II) complexes with five different ligands (2,3,4-paoH, dapdoH2, and LH4). The authors synthesized these complexes and characterized them with FTIR, NMR, and XRD. These compounds are in fact very interesting.  I am in agreement with the authors however, I have a few comments and suggestions that will hopefully improve the manuscript if considered by the authors.

We thank the referee for her/his warm comments.

  1. The authors wrote that they checked the purity of compounds with 1H NMR however, some of their complexes 2, 3, and 4 were decomposed in the D6-DMSO. It is then unclear how they check the purity of complexes, 2, 3, and 4? I suggest authors to provide values of %purity of their compounds in the experimental section. Moreover, in lines 260-261, they wrote …’The spectra do not exhibit bands that are present in the free ligands 2paoH, 3paoH, and 4-paoH, dapdoH2, and LH4, suggesting their purity’…. Authors need to clarify their how FTIR spectra confirms purity?

Concerning the first point, in part 3.1 we meant that the purity of the free organic ligands (and not the purity of the complexes) had been checked with 1H NMR spectroscopy; this is a common practice in our lab, especially for known ligands (such as dapdoH2 and LH4) that have been synthesized before by other research groups, but also for organic ligands (such as 2paoH, 3paoH and 4paoH) we purchase from commercial sources. To make clear this point, we have rephrased the relevant sentence in the first paragraph of part 3.1. Concerning the second point, what we mean that if the IR and Raman spectra of the complexes contain bands/peaks at exactly the same wavenumbers with those of the free ligands, this will be an indication of contamination (i.e., impurity) of the complexes with amounts of the free ligands. This point is better clarified now in the second sentence of the first paragraph of part 3.2.

  1. It is hard to follow what authors wants to convey in lines 213 - 214, ‘The use of 1:1 reaction ratio in the case of the CdI2/(2,3,4-paoH) systems does not affect the product identity’. Can they explain what is the product identity? In the same section, in lines 219 - 220, they wrote that the ratio of 2:1 dapdoH2:CdI2) was used while Figure 2 displayed 1:1 ratio between dapdoH2:CdI2 for formation of complex 4. I suggest authors to clarify this and write equivalent of CdI2 used in the Figure 2, perhaps below CdI2.

Concerning the first point: The complexes 1, 2 and 3 were prepared (see part 3.2) by using an 1:2 CdI2: (2,3,4-paoH) experimental reaction ratio. In part 2.1, we state that the same complexes can be obtained even by using an 1:1 ratio. Concerning the second point: Complex 4 was prepared (see part 3.2) by using an 1:1 CdI2:dapdoH2 experimental reaction ratio. In part 2.1, we state that the same complex can be also obtained by using an excess of dapdoH2, i.e., CdI2:dapdoH2 = 1:2. We have made both points clear in the revised version. As far as Figure 2 is concerned, it is widely recognized that by writing “CdI2” we mean 1 equivalent.

  1. In lines, 292 - 294, it is unclear which band are they talking. I suggest authors to write band value in cm-1 and explain how much in numbers cm-1 it is decreased.

The comment is absolutely correct, and we have followed the referee’s valuable instruction.

  1. Figure 3 has very poor resolution. I suggest authors to increase the resolution and the font size in Figure 3.

The resolution and the font size in Figure 3 have been improved as correctly indicated by the referee.

  1. In Figure 4, I suggest authors to add FT-Raman spectra of free ligand 3paoH so it is easier for readers to see how FT-Raman spectra changes from non-complexed to the complexed structure.

We have added the Raman spectrum of the free ligand 3paoH in the revised Figure 4. As correctly stated by the referee, this new figure is much more helpful for the reader of the paper.

  1. In the XRD structural characterization section, as the structure of compounds 2 and 3 are very similar so I suggest authors to compress their discussion and explain them together.

The comment is absolutely correct and we have in part followed the referee’s instruction. Instead of combining the two structural discussions into one, we have condensed the description of the structure of complex 4. Due to the different Wyckoff positions and the different y points where the mirror plane cuts the b axis, a common description would be confusing for the non-familiar readers.

  1. In line 188 of experimental section, I suggest authors to give average yield of complex 2 instead of range (60 – 70%) which is currently given.

We give the average yield, as suggested by the reviewer.

  1. Moreover, conclusion section need to be revised thoroughly. It should be described briefly and scientifically.

We do believe that the original conclusion section was brief and scientifically sound, and needed no revision. It consisted of the main conclusions from the chemical and technological viewpoints. However, in order to prove that we respect the referee’s opinion, we have removed the last paragraph of this section which described our ongoing research efforts.

  1. The paper is very tedious to read, sometimes non-scientific language is used at several places, and extensive editing of English language and style is recommended. At some places conclusions are merely based on assumptions and not supported by the results.

If we see the introduction of the report by the reviewer, this comment is a big surprise for us! According to our opinion, the conclusions are sound and not “merely based on assumptions” and they are fully supported by the results. The term “tedious” is subjective; although we respect the referee’s comment, we do believe that it is not tedious! Although our mother language is not English, the ms has been checked by a colleague who worked in a UK university and found the English language more than satisfactory. Concerning the “style” of the ms, at the age of 70, I have written more than 400 papers which have been published in peer-reviewed journals, and I am convinced that the terminology used and the style of writing is well above those of the papers that appear in the Inorganic Chemistry section of MOLECULES. I hope that the changes made in the ms through the excellent work of the reviewers (including this reviewer) have improved it a lot.

  1. Few examples are identified here but Language needs to extensively improved, Line 133, ‘The polish group’. I suggest authors to replace it with the name of authors.

The suggestion is correct and we have followed it.

  1. Line 211, …’Figure 2 characteristic deserve synthetic comments’….

We have modified this sentence.

  1. Line 230, A peculiar event took place…

We have modified this sentence.

  1. Line 232, pink-pale violet! I suggest authors to describe this color in a better way so it is easier for readers to follow.

The comment is correct. We have used the term “pale violet” in the revised ms.

  1. Lines 311-313, 345-347 needs to written more precisely so it is easier to follow.

We have re-written these sentences, so their meaning is easier to follow.

  1. Line 145 in the experimental section, ‘synthetic manipulations’….

We have replaced the term “synthetic manipulations” by “experimental manipulations”.

  1. Line 264, conclusion section, welcome case in the area….

We have replaced the word “case” by the word “example” which describes with an unambiguous way the meaning.

Reviewer 2 Report

The reviewed manuscript has practical relevance, yielding interesting applications, worth publishing, once the authors address some aspects, which in my opinion require their attention. Although it is overall well written and argued (the corresponding authors are well published), there are some parts of the manuscript which in my opinion require attention.

The abstract is far too detailed and lengthy for the intended purpose. The same goes for the introduction section as well.       

Keywords are used to assist search engines to find the article, as such, is seems redundant to use keywords apparent in the title.

Page 2, Line 62, the authors refer to “heavy toxic metal ions” and shortly after state that “toxic metals” are neither biodegradable nor decomposable, which of course is correct. However the transition from ions to metals is confusing, as the initially referred-to ions have generally a short life spam and are rendered inert after a period of action.

Page 5, lines 198-201. I think this would be better suited for the Experimental section.

Author Response

Reviewer # 2

The reviewed manuscript has practical relevance, yielding interesting applications, worth publishing, once the authors address some aspects, which in my opinion require their attention. Although it is overall well written and argued (the corresponding authors are well published), there are some parts of the manuscript which in my opinion require attention.

We thank the referee for her/his warm general comment.

  1. The abstract is far too detailed and lengthy for the intended purpose. The same goes for the introduction section as well.

We agree with Reviewer #2. We have significantly condensed the abstract and the introduction.

  1. Keywords are used to assist search engines to find the article, as such, is seems redundant to use keywords apparent in the title.

We have removed the keyword “solvent extraction” because this appears in the title.

  1. Page 2, Line 62, the authors refer to “heavy toxic metal ions” and shortly after state that “toxic metals” are neither biodegradable nor decomposable, which of course is correct. However the transition from ions to metals is confusing, as the initially referred-to ions have generally a short life spam and are rendered inert after a period of action.

The comment is absolutely correct. We have used the term “toxic metals” in both cases. Also we agree that the transition from ions to metal was confusing. We have clarified this point by adding the sentence “Its ion has generally a short life span and is rendered inert after a period of action” which was proposed by the reviewer.

  1. Page 5, lines 198-201. I think this would be better suited for the Experimental section.

We completely agree with the opinion of the referee. Thus, we have transferred the relevant information from part 2.1 (“Comments on the Syntheses of the Complexes”) to part 3.1 (“Materials and Spectroscopic-Physical Measurements”).

Reviewer 3 Report

This manuscript by Routzomani et al. reports the molecular basis of the solvent extraction of cadmium(II) using 2-pyridyl oximes through a synthetic inorganic chemistry approach and a proposal for more efficient extractants. This is a comprehensive study, I suggest a minor revision. My comments are as follows:

  1. For clarity, I suggest authors to add a scheme describing the main content of this study at the beginning of the main text.
  2. More information on current solvent extraction methods should be added in INTRODUCTION.
  3. Can authors provide some additional theoretical calculation (e.g., DFT) to explain the mechanism in Figure 3.
  4. English should be polished before publication.

Author Response

Reviewer # 3

This manuscript by Routzomani et al. reports the molecular basis of the solvent extraction of cadmium(II) using 2-pyridyl oximes through a synthetic inorganic chemistry approach and a proposal for more efficient extractants. This is a comprehensive study, I suggest a minor revision. My comments are as follows:

We thank the referee for her/his very positive general comment.

  1. For clarity, I suggest authors to add a scheme describing the main content of this study at the beginning of the main text.

The comment is correct. However, instead of adding a scheme in the main text, we believe it will be better to illustrate this information in the Graphical Abstract by creating a new, informative one. This new material describes the main content of our study.

  1. More information on current solvent extraction methods should be added in INTRODUCTION.

We completely agree. We have included new information in the first paragraph of section 1 (“Introduction”) and describe two additional extraction methods currently employed.

  1. Can authors provide some additional theoretical calculation (e.g., DFT) to explain the mechanism in Figure 3.

This idea is very good. However, this is impossible for the moment mainly due to time limitation. This is an experimental study and the proposed mechanism is based on known reactivity patterns of the amidoxime group (although the entire transformation is observed for the first time). We are asking your and referee’s indulgence to avoid DFT calculation at the present stage of this work.

  1. English should be polished before publication.

As suggested, English have been polished.

Reviewer 4 Report

The paper presents an interesting study concerning the solvent extraction of toxic Cd(II) using 2-pyridyl oximes as extractants. The removal of toxic metals is an important research topic and its further exploration is perfectly justified. 

The authors present a well structured and coherent work plan exploring different extractants and ligands by a wide set of experimental methodologies namely single crystal X-ray crystallography and spectroscopic techniques. The results are properly discussed and promising conclusions have been obtained.

I believe this work is worthy to be published in Molecules journal. I would like to suggest, if possible, to improve the quality/resolution of some figures (namely Figures 3, S1, S4, S5, S7 and S8) – the current quality hampers a proper analysis by the readers. In line 192 (page 5), there is a small typo: “continiuation” should be replaced by “continuation”.

Author Response

Reviewer # 4

The paper presents an interesting study concerning the solvent extraction of toxic Cd(II) using 2-pyridyl oximes as extractants. The removal of toxic metals is an important research topic and its further exploration is perfectly justified. 

The authors present a well structured and coherent work plan exploring different extractants and ligands by a wide set of experimental methodologies namely single crystal X-ray crystallography and spectroscopic techniques. The results are properly discussed and promising conclusions have been obtained. I believe this work is worthy to be published in Molecules journal.

We thank the referee for her/his very warm and positive comments.

  1. I would like to suggest, if possible, to improve the quality/resolution of some figures (namely Figures 3, S1, S4, S5, S7 and S8) – the current quality hampers a proper analysis by the readers.

The suggestion is absolutely logical. We have improved the quality/resolution of Figure 3 in the main body of the article (please, see also our response to point 4 raised by Reviewer #1). Concerning Figures S1, S4, S5, S7 and S8, it was impossible to find the original files in order to improve their quality (MSc student A.R., who has left the lab and is working now in a pharmaceutical company could not find them). However, the general appearance of the figures-which are all in the “Supplementary Materials” section- is not so bad and the reader can see the spectra and read the wavenumbers/chemical shifts; moreover, these values appear in part 3.2 of the “Experimental Section”.

  1. In line 192 (page 5), there is a small typo: “continiuation” should be replaced by “continuation”.

The mistake has been corrected.